# Seroepidemiology of Tetanus among Healthy People Aged 1–59 Years Old in Shaanxi Province, China

**DOI:** 10.3390/vaccines10111806

**Published:** 2022-10-26

**Authors:** Chao Zhang, Weijun Hu, Yu Ma, Li Li, Yuan Si, Shaobai Zhang

**Affiliations:** Department of Immunization Program, Shaanxi Provincial Center for Disease Control and Prevention, Xi’an 710054, China

**Keywords:** tetanus, antibody level, seroprevalence

## Abstract

The study aims to determine the seroprevalence of antibodies against tetanus among healthy people aged 1–59 years old in Shaanxi province. IgG against tetanus in serum samples were detected by ELISA. 6,439 subjects were enrolled. The positive rate (≥0.01 IU/mL) was 84.39% and GMC was 0.03 IU/mL. There were significant differences in positive rates (*χ^2^* = 308.944, *p* < 0.01) and GMC (Z = 5,200,000, *p* < 0.01) among different age groups. The positive rates (*χ^2^* = 304.3, *p* < 0.01) and GMCs (*χ^2^* = 146.417, *p* < 0.01) showed regional differences. Both full protection rate (≥0.1 IU/mL) (*χ^2^* = 36.834, *p* < 0.01) and GMC (Z = 688,000, *p* < 0.01) increased with the doses of tetanus-toxoid-containing vaccines (TTCVs) administered. The positive rate (*χ^2^* = 54.136, *p* < 0.01) and GMC (Z = 140,200, *p* < 0.01) decreased gradually with the time interval after full immunization with TTCVs. The full protection rate (≥0.1 IU/mL) (*χ^2^* = 176.201, *p* < 0.01) and GMC (Z = 629,900, *p* < 0.01) decreased with the interval (years) since the last dose of TTCVs. There were significant differences in the positive rates and GMCs for different ages, regions, immunization histories of TTCVs, and doses of TTCVs administered. The full protection rate and GMC decreased with the interval following full immunization with TTCVs and the interval since the last dose of TTCVs. The importance of using tetanus booster doses should be emphasized in adolescents and adults.

## 1. Introduction

Tetanus is a rare but fatal disease that affects the nervous system. It is caused by a neurotoxin produced by Clostridium tetani, a gram-positive anaerobic bacteria that is widely present in the surrounding environment, particularly in dirt, soil and dust [1]. Deep wounds with lacerated and bruised margins, devitalized tissue, and soil contamination are at high risk of tetanus [2,3]. 

Although tetanus is rare in developed settings, it remains common in many developing countries, and still presents huge diagnostic and therapeutic challenges. Tetanus can occur at any age, but mainly occurs among newborns and women with unclean childbirth and poor postpartum health conditions [4]. It was estimated that about 34,000 neonates died of neonatal tetanus (NT) in 2015 worldwide [4,5]. In China, a total of 3992 NT cases were reported from 2010 to 2017, for an average incidence of 3.2 per 100,000 [6].

Immunization is the most effective and reliable strategy for preventing tetanus incidence. By 2018, 86% of infants worldwide (116.3 million) had received three doses of Diphtheria-Tetanus-Pertussis (DTP) vaccine [7]. The World Health Organization (WHO) estimated that the number of neonates who died of NT had dropped by 96% in 2015 compared with those in 1988, due to tetanus immunization [5]. In China, primary vaccination for infants and young children includes three doses of DTP at three, four and five months old followed by a fourth dose at 18 to 24 months old, with one Diphtheria-Tetanus (DT) booster dose recommended at six years old [8]. Since DTP was introduced into the immunization program in China in 1978, the vaccination rate of four doses of DTP for Chinese children exceeded 99% by 2011, and the tetanus incidence showed a significant downward trend [9]. The NT incidence in China had dramatically dropped from 10 cases per 100,000 in 2008 to only one case per 100,000 in 2017 [6,10]. There is no doubt that tetanus-toxoid-containing vaccines (TTCVs) have played a vital role in reducing tetanus incidence, whether around the world or in China alone. However, healthy people still face the threat of tetanus; there are still a certain number of tetanus cases reported, even in some developed countries that have high rates of immunization [11,12]. A total of 594 tetanus cases were reported in Italy from 2001 to 2010, with an average annual incidence of 1.0/1,000,000 population [13]. A special survey for construction workers showed that although Italy as a whole had a high TTCVs coverage, the immunization status of special populations, such as construction workers, was not very satisfactory [14]. Many more cases are reported in Japan than in other developed countries [12]. Sero-surveillance is an important tool for monitoring vaccine-preventable diseases (VPDs), and IgG antibodies against tetanus are one of the indicators that can help with monitoring the effectiveness of TTCVs in vaccinated people. Therefore, it is important to evaluate antibody levels of tetanus by detecting IgG antibodies against tetanus in healthy populations.

In China, studies showed that the positive rate of tetanus antibodies had geographical differences and varied from region to region, and the positive rate of tetanus antibodies ranged between 74.85% and 83.67% [15,16,17,18]. However, the seroepidemiology of tetanus remains unclear in many parts of China. As far as we know, there are currently no studies on tetanus antibody levels and its persistence through such large-scale populations in China. This study aimed to study the seroprevalence of antibodies against tetanus among healthy people in Shaanxi province in Northwest China, which is economically underdeveloped, and to provide data support and a theoretical basis for adjusting vaccination strategies.

## 2. Materials and Methods

### 2.1. Subjects and Study Design

A large-scale epidemiological investigation of hepatitis B virus (HBV) was conducted in Shaanxi province (with a population of 38.35 million) by the Center for Disease Control and Prevention (CDC) in 2017. According to the HBV Seroepidemiological Survey Project Program in Shaanxi Province in 2017, a multi-stage stratified random sampling method was used to select healthy people aged 1–59 years old from 30 county-level settings in 10 cities in Shaanxi province in 2017 (Figure 1). The required sample size was calculated by the prevalence of HBsAg in people aged 1–14 years old, 15–29 years old and 30–59 years old, respectively, in 2006. The minimum sample size required for the 1–14, 15–29 and 30–59 age groups was 1902, 2212 and 2737, respectively, and finally, nearly 7000 serum samples were collected from individuals aged 1–59 years who were residents in this district (residence ≥six months) for the HBV epidemiological investigation. In this study, to get a large enough sample, serums with a volume of more than 100 μL were selected. All the serum samples used for the HBV serosurvey were stored at −20 °C before analysis.

### 2.2. Data Collection

Basic demographic and epidemiological information, such as sex, age and immunization history were obtained by face-to-face interviews with structured questionnaires by the local CDC. EpiData 3.1 (EpiData Association, Odense, Denmark) was used for double data entry and consistency checks.

### 2.3. Laboratory Methods

Levels of IgG against tetanus were quantitatively measured using commercial enzyme-linked immunosorbent assay (ELISA) kits (Virion/Serion GmbH, Würzburg, Germany) according to the manufacturer’s instructions, and antibody activity is expressed in international units (IU)/mL. A level of tetanus antibody <0.01 IU/mL was defined as ‘no immune protection or seronegativity’; an antibody level between 0.01 and 0.1 IU/mL was defined as ‘basic protection or low seropositivity’; and ≥0.1 IU/mL was defined as ‘full protection’. A subject was considered as positive with an antibody level ≥0.01 IU/mL, which included both the low seropositivity (≥0.01 IU/mL) and seropositivity categories above (≥0.1 IU/mL).

### 2.4. Statistical Analysis

Levels of IgG against tetanus were summarized as geometric mean concentration (GMC) and the prevalence of seropositivity of antibodies was calculated as a percentage. If the antibody results were non-normally distributed, non-parametric test methods were used. The GMC of antibodies were compared among groups using Kruskal–Wallis test or Jonckheere–Terpstra test, and seropositivity was compared between subgroups using chi-squared tests. Linear-by-Linear Association was used to test the change trend. SPSS software (version 25.0; SPSS Inc., Chicago, IL, USA) was used for data analysis and a *p*-value of <0.05 was considered significant.

### 2.5. Ethical Approval

This study received ethical approval from the Ethics Committee of Shaanxi Provincial Center for Disease Control and Prevention (No. 2017-MY01). All experiments performed in this study were in accordance with the national laws and regulations of China.

## 3. Results

### 3.1. SubsectionSociodemographic Characteristics of Study Population

6439 participants aged 1–59 years old were included in this study, including 1092 in northern Shaanxi (Yan’an and Yulin), 4031 in central Shaanxi (Xi’an, Tongchuan, Baoji, Xianyang and Weinan) and 1316 in southern Shaanxi (Ankang, Hanzhong and Shangluo). The Han ethnic group accounted for 99.8% and the others 0.2%. In 4307 subjects over 18 years old, on occupational distribution, farmers, workers, cadres or staff, students, teachers, medical staff, public service personnel and others accounted for 71.79%, 5.02%, 5.18%, 3.99%, 1.18%, 3.09%, 2.55% and 7.20%, respectively; on education level, illiterate, primary school, junior high school, high school, college and above and unknown accounted for 4.41%, 16.46%, 43.91%, 20.48%, 13.58% and 1.16%, respectively. The median age was 27 (13–44) years old. The male-to-female ratio was 1:1.1 (3017:3422).

### 3.2. Comparison among Subjects of Different Ages

The positive rates of antibodies against tetanus in different age groups ranged from 73.43% (50 to 59 years old) to 97.47% (seven to nine years old) (*χ^2^* = 308.944, *p* < 0.01) and their antibody GMCs ranged from 0.02 IU/mL (50 to 59 years old) to 0.16 IU/mL (one to two years old) (Z = 5,200,000, *p* < 0.01) (Table 1). The highest full protection rate was in the six years old group (69.51%), and the lowest was in the 50 to 59 years old group (1.83%) (Figure 2A).

### 3.3. Distribution among Recruitment Regions

The positive rates of antibodies against tetanus among 10 cities ranged from 74.34% (Xi’an) to 95.52% (Yulin) (*χ*^2^ = 304.3, *p* < 0.01) and the antibody GMCs ranged from 0.028 IU/mL (Yan’an) to 0.05 IU/mL (Shangluo) (*χ*^2^ = 146.417, *p* < 0.01) (Table 2).

### 3.4. Immunization History of TTCVs

32.29% (2079/6439) of subjects had a clear TTCVs immunization history, 38.16% (2457/6439) of subjects had no TTCVs immunization history and 29.55% (1903/6439) of subjects’ immunization history was unknown. There were significant differences in positive rates (*χ*^2^ = 241.8, *p* < 0.01) and GMCs (*χ*^2^ = 1641.475, *p* < 0.01) among different groups of immunization history (Table 3).

### 3.5. Comparison of Different Doses

Of the 2079 subjects with an immunization history of TTCVs, 2064 had a definite number of doses. Both full protection rate (≥0.1 IU/mL) (*χ*^2^ = 36.834, *p* < 0.01) and GMC (Z = 688,000, *p* < 0.01) increased with the doses of TTCVs received. The full protection rate of the highest group was 51.49% (five doses), the lowest was 12.50% (one dose) and the highest was more than four times than the lowest. At the same time, the antibody GMCs ranged from 0.03 IU/mL (one dose) to 0.10 IU/mL (five doses) (Z = 688,000, *p* < 0.01), and the highest was more than three times than the lowest (Table 4). 

### 3.6. Antibody Level following Basic and Booster Immunization of TTCVs

Of the 1039 subjects who completed basic (four doses of DTP) and booster (one dose of DT) immunization (full immunization), the vaccination time of 990 subjects were clearly recorded. The positive rate (*χ*^2^ = 54.136, *p* < 0.01) and GMC (Z = 140,200, *p* < 0.01) decreased gradually with the time interval after the full immunization. After 10 years of full immunization, the positive rate and full protection rate decreased from 97.76% and 78.36% within 1 year to 81.25% and 21.25% over 10 years, respectively, which was only 83.11% and 27.12% of that within 1 year, respectively, and the GMC decreased from 0.31 IU/mL to 0.03 IU/mL (Table 5) (Figure 2B).

### 3.7. Antibody Level According to the Interval since the Last Dose of TTCVs

Of the 2079 subjects who had received TTCVs, the vaccination time of the last dose was clearly recorded in 1,976 subjects. The full protection rate (≥0.1 IU/mL) (*χ*^2^ = 176.201, *p* < 0.01) and GMC (Z = 629,900, *p* < 0.01) also decreased gradually with the time interval since the last dose. The full protection rate decreased from 71.48% within 1 year to 26.58% over 10 years and 10.61% over 20 years, which was only 37.19% and 14.84% of that within 1 year, respectively, and the GMC decreased from 0.23 IU/mL within 1 year to 0.04 IU/mL over 10 years and 0.03 IU/mL over 20 years (Table 6) (Figure 2C).

## 4. Discussion

It was found that in this study, the positive rate of IgG against tetanus in healthy people in Shaanxi province (84.39%) was not significantly different from that in developed countries such as Singapore, Italy and Korea [19,20,21]. However, it was higher than that in a few developing countries including Turkey, Congo, and Uganda [22,23,24]. Overall, it was confirmed that the TTCVs and vaccination procedures used in China were effective. Although a few studies had been carried out in some provinces of China, some of which were limited by sample size and therefore cannot conclusively demonstrate problems with tetanus immunization in some districts, the findings still indicate gaps in tetanus immunization among different provinces. In studies conducted in China, the positive rates and GMCs of tetanus antibodies in healthy people varied from region to region, ranging from 74.85% to 83.67%. Tong et al. reported that the positive rate and GMC of tetanus antibodies were, respectively, 74.85% and 0.05 IU/mL among 680 recruits recruited from 12 provinces, and the positive rates ranged from 47.62% to 100%, the GMCs ranged from 0.02 IU/mL to 0.09 IU/mL [15]. Although subjects in Tong’s research had a wide geographical scope, the research was limited by age and sample size. Wu et al. found that the positive rate of tetanus antibodies was 82.11% and GMC was 0.063 IU/mL among 587 healthy subjects aged 2–17 years in Hebei province [16]. Xu et al. reported that the positive rate of tetanus antibodies was 76.02% and GMC was 1.7 IU/mL through a large sample of more than 3000 subjects aged 0–≥40 years old in Hangzhou of Zhejiang province [17]. Liu et al. found that the positive rate of tetanus antibodies was 83.67% and the GMC was 0.099 IU/mL among 1482 subjects aged 0–≥50 years old from eight counties in Henan province [18]. The positive rate in Liu’s research was consistent with that in this study (84.39%). On one hand, the reasons for the inconsistency above may be related to the difference in survey methods and laboratory tests. On the other hand, the inconsistency may be related to a number of factors such as ethnicity, age structure, districts and investigation time, indicating that there are certain differences in the positive rates and GMCs of tetanus antibodies among healthy people in different areas. To date and to the best of our knowledge, there are currently no studies on the seroepidemiology of tetanus antibodies among healthy people based on such a large population size in all cities all over the province, as this study in China. In this study, there were significant differences in the positive rates and GMCs of tetanus antibodies among subjects in different cities (*p* < 0.05). Of which, the highest positive rate was 95.52% in Yulin and the lowest was only 74.34% in Xi’an. The highest GMC was 0.05 IU/mL in Shangluo and the lowest was only 0.028 IU/mL in Yan’an. The results above showed that there were differences not only among different provinces, but also within provinces, which further confirmed the necessity of carrying out seroepidemiology in this region. Despite the use of the same vaccines and the same vaccination procedures in the same province, there were large differences. The reasons for the differences need to be further explored and follow-up targeted thematic surveys should be carried out. More detailed vaccination strategies need to be developed based on the specific situation and survey results of each city in this study.

A number of studies have shown that despite the completion of full immunization in childhood (five doses of TTCVs), tetanus antibody levels in the body will gradually decrease with age [21,25,26]. Zhang et al. reported that the positive rate of tetanus antibodies was only 31.3% in ≥50-year-old group compared with 80.2% in <one-year-old group [27]. In this study, the positive rate of tetanus antibodies decreased from 97.35% in the 1 to 2 years old group to 74.43% in the 50 to 59 years old group, and the GMC also dropped from 0.16 IU/mL to 0.02 IU/mL. Some subjects can obtain lifelong immunity to tetanus by vaccination, but most people can only maintain effective antibody levels against tetanus infection for 10 years after vaccination due to waning immunity, and the adult Td or Tdap vaccine is recommended every 10 years with complete prior immunization [28]. By contrast, the full protection rate fell more sharply, from 64.60% in the 7 to 9 years old group to 1.83% in the 50 to 59 years old group. Since one dose of DT booster was administered at six years old, the antibody reached the highest level and then gradually decreased with age. The IgG against tetanus declined with age (*p* < 0.01) [21]. Some studies reported that tetanus and diphtheria antibody levels decreased with age following the final vaccination, and the estimated half-life of antibodies was 11 years [25,29]. In addition, since 1978, DPT and DT were gradually included in the planned immunization in China, that is, the subjects over 40 years old in this study might not be vaccinated against DPT or DT. These data explain why most elderly adults lack protective antibodies. In this study, the positive rate and antibody level decreased gradually with the time interval after the full immunization (five doses of TTCVs). After 10 years of full immunization, the positive rate and GMC of tetanus antibodies decreased to the lowest levels; 81.25% and 0.03 IU/mL, respectively. Liu et al. found that the positive rate of tetanus antibodies decreased from 98.39% one year after the last dose to 81.08% eleven years after the last dose, and the GMC dropped from 0.129 IU/mL to 0.033 IU/mL [18]. Liu’s results are basically consistent with the results of this study. A large number of studies have reported that the positive rates and GMCs of tetanus antibodies decrease with age, which is essentially caused by the prolonged time interval after vaccination [17,25,30]. In reality, the protection rate of tetanus antibodies among adults is generally low; a few countries have recognized this problem and have actively developed appropriate booster immunization programs for adults.

Although there is no tetanus vaccine booster program for adults in China, relevant studies have been conducted in special populations. Farmers, construction workers, police and soldiers are high-risk occupational groups for tetanus infection. Soldiers are responsible for training, combat and other tasks, including flood disaster and earthquake disaster rescue, and they are vulnerable to scratches, crushing and other traumas. They often have open, unclean wounds and cannot receive timely medical care, which may increase the risk of Clostridium tetani infection. 74.85% of recruits in a military unit were positive for tetanus antibodies, while a quarter still lacked protective antibodies against tetanus, which was not enough for such a high-risk population. The data showed that there were significant differences in the positive rate of tetanus antibodies and GMCs between peacekeeping officers and recruits. One year after the peacekeeping officers were vaccinated, the positive rate of tetanus antibodies was 100%, with an increase of 25.15% compared with the unvaccinated recruits, and the GMC was also greatly improved, which was 3.94 times that of the unvaccinated recruits. The data showed that both the positive rate and GMC of tetanus antibodies were significantly improved following booster immunization, and the effect was obvious. In the U.S. military, a current tetanus, diphtheria and acellular pertussis (Tdap) vaccination status is compulsory for service [31]. 

According to the WHO, 10,301 cases of tetanus were reported worldwide, of which 3551 were neonatal in 2015 [4]. In China, a total of 3992 NT cases were reported during 2010 to 2017, for an average incidence of 0.032‰ [6]. Although the annual incidence decreased from 0.058‰ in 2010 to 0.0059‰ in 2017, there were still 93 NT cases reported in 2017. Reducing the morbidity and mortality caused by tetanus, especially maternal and neonatal, is one of the major targets of health organizations worldwide. In response to an estimated 6.7 neonatal deaths per 1000 live births due caused by tetanus, the World Health Assembly launched the maternal and neonatal elimination initiative in 1988 [2]. The program includes three principal parts: clean childbirth, enhanced surveillance, and vaccination [2]. Clean childbirth and enhanced surveillance are no longer an issue in China. In contrast, tetanus booster immunizations for adults and women of childbearing age were not included in the immunization program. NT is prevented by maternal immunization, and it is estimated that 84% of newborns were protected from tetanus via maternal vaccination. For women who are not vaccinated or have never been vaccinated against tetanus, two doses of tetanus toxoid are recommended four weeks apart during pregnancy. This program should provide adequate antibody protection for newborns. For durable maternal protection, five doses should be given; the third dose should be given six months after the second dose, and two subsequent doses should be given five years and ten years later [2]. Tdap was approved for adults in USA in 2005 and recommended for every pregnancy in 2012, with an optimal timing of 27 to 36 weeks of pregnancy [32], and has proven that exposure to Tdap during pregnancy is safe, including during early pregnancy (0 to 13 weeks of pregnancy) [31]. In the United States, Tdap and Td coverage among adults (≥18 years) were 28.9% and 57.5%, respectively [33], and the coverage in pregnant women was 41.7% in 2013 [34]. In contrast, tetanus booster vaccinations for adolescents or adults have not yet been adopted in China. In this study, up to 16% of women of childbearing age (20–39 years old) did not reach protective antibody levels, and they or their children may be at risk of tetanus infection. A booster immunization with the sixth dose TTCVs in adults should be considered.

## 5. Conclusions

There were significant differences in the positive rates and GMCs of IgG against tetanus among different ages, regions, immunization histories of TTCVs, and doses of TTCVs inoculated. Full protection rate and GMC decreased with the interval (years) following full immunization of TTCVs and the interval since the last dose of TTCVs. Besides basic immunization during childhood, the importance of using tetanus booster doses should be emphasized in adolescents and adults; it is necessary to take corresponding immunization measures for special groups, such as women of childbearing age and soldiers. To further explore the reasons for the differences among different cities, follow-up targeted thematic surveys should be conducted.

## Figures and Tables

**Figure 1 vaccines-10-01806-f001:**
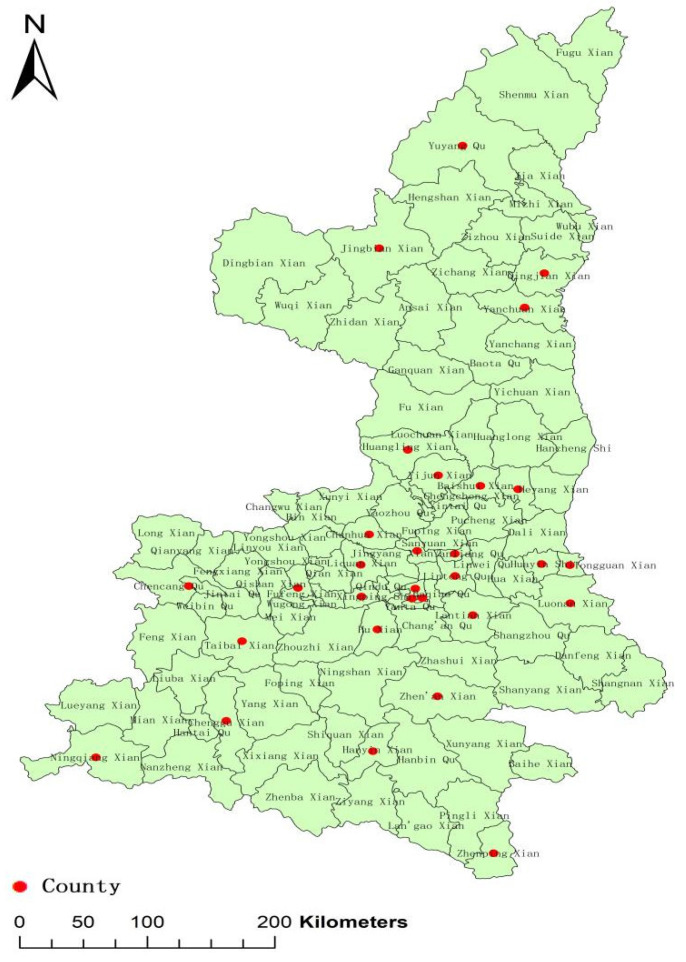
Distribution of 30 county-level settings selected from 10 cities in Shaanxi province.

**Figure 2 vaccines-10-01806-f002:**
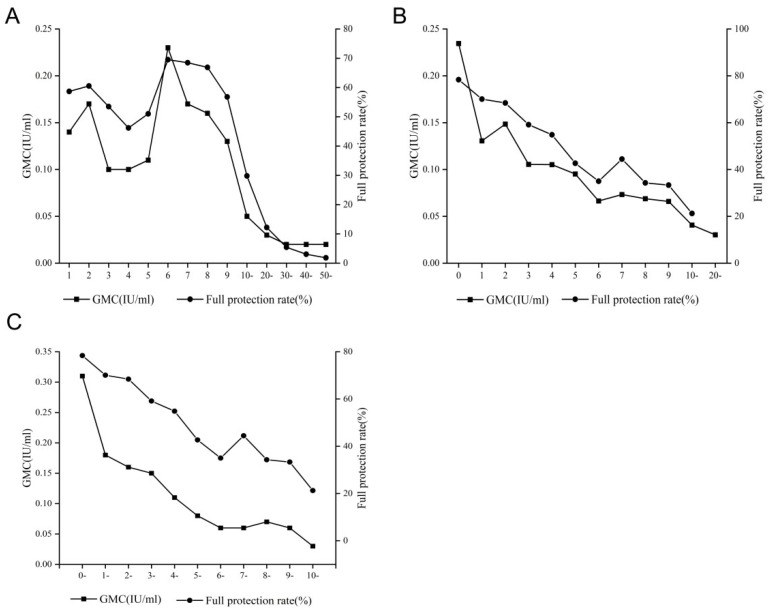
Full protection rate (≥0.1 IU/mL) and GMC according to age (**A**), interval (years) following basic and booster immunization of TTCVs (**B**), and the interval (years) since the last dose (**C**).

**Table 1 vaccines-10-01806-t001:** Seroprevalence and tetanus antibody level according to age.

Subgroup		*n*	GMC [IU/mL, (95% CI)]	*Z, p*	Proportion [*n* (%)]			Positive Rate (≥0.01 IU/mL) [*n* (%)]	*χ^2^, p*
					<0.01 IU/mL	0.01–0.1 IU/mL	≥0.1 IU/mL		
Age(years)	1–2	189	0.16	5,200,000 *, <0.01	5(2.65)	71(37.57)	113(59.79)	184(97.35)	308.944 ^#^, <0.01
	3–4	274	0.10		11(4.01)	127(46.35)	136(49.64)	263(95.99)	
	5–6	309	0.16		8(2.59)	113(36.57)	188(60.84)	301(97.41)	
	7–9	435	0.16		11(2.53)	143(32.87)	281(64.60)	424(97.47)	
	10–19	1080	0.05		114(10.56)	644(59.63)	322(29.81)	966(89.44)	
	20–29	1393	0.03		198(14.21)	1025(73.58)	170(12.20)	1195(85.79)	
	30–39	755	0.02		146(19.34)	568(75.23)	41(5.43)	609(80.66)	
	40–49	1018	0.02		250(24.56)	737(72.40)	31(3.05)	768(75.44)	
	50–59	986	0.02		262(26.57)	706(71.60)	18(1.83)	724(73.43)	
Total		6439	0.03		1005(15.61)	4134(64.20)	1300(20.19)	5434(84.39)	

* Jonckheere–Terpstra test; # Linear-by-Linear Association; GMC: geometric mean concentration.

**Table 2 vaccines-10-01806-t002:** Seroprevalence and tetanus antibody level in different cities.

City	*n*	GMC [IU/mL, (95% CI)]	*χ*^2^, *p*	Proportion [*n* (%)]			Positive Rate (≥0.01 IU/mL) [*n* (%)]	*χ*^2^, *p*
				<0.01 IU/mL	0.01–0.1 IU/mL	≥0.1 IU/mL		
Xi’an	1356	0.03	146.417 *, <0.01	348 (25.66)	712 (52.51)	296 (21.83)	1008 (74.34)	304.3 #, <0.01
Tongchuan	226	0.03		49 (21.68)	129 (57.08)	48 (21.24)	177 (78.32)	
Baoji	677	0.03		160 (23.63)	366 (54.06)	151 (22.30)	517 (76.37)	
Xianyang	889	0.03		146 (16.42)	589 (66.25)	154 (17.32)	743 (83.58)	
Weinan	883	0.04		80 (9.06)	632 (71.57)	171 (19.37)	803 (90.94)	
Yan’an	423	0.03		86 (20.33)	277 (65.48)	60 (14.18)	337 (79.67)	
Hanzhong	455	0.04		45 (9.89)	316 (69.45)	94 (20.66)	410 (90.11)	
Yulin	669	0.04		30 (4.48)	502 (75.04)	137 (20.48)	639 (95.52)	
Ankang	454	0.04		41 (9.03)	329 (72.47)	84 (18.50)	413 (90.97)	
Shangluo	407	0.05		20 (4.91)	282 (69.29)	105 (25.80)	387 (95.09)	
Total	6439	0.03		1005 (15.61)	4134 (64.20)	1300 (20.19)	5434 (84.39)	

* Kruskal–Wallis test; #Pearson’s chi-squared test; GMC: geometric mean concentration.

**Table 3 vaccines-10-01806-t003:** The immunization history of tetanus-toxoid-containing vaccines (TTCVs) among 6439 subjects enrolled in this study.

Subgroup		*n*	GMC [IU/mL, (95% CI)]	*χ*^2^, *p*	Proportion [*n* (%)]			Positive Rate (≥0.01 IU/mL) [*n* (%)]	*χ*^2^, *p*
					<0.01 IU/mL	0.01–0.1 IU/mL	≥0.1 IU/mL		
Immunization history	Yes	2079	0.09	1641.475 *, <0.01	137 (6.59)	940 (45.21)	1002 (48.20)	1942 (93.41)	241.8 #, <0.01
	No	2457	0.02		575 (23.40)	1799 (73.22)	83 (3.38)	1882 (76.60)	
	Unknown	1903	0.03		293 (15.40)	1395 (73.31)	215 (11.30)	1610 (84.60)	
Total		6439	0.03		1005 (15.61)	4134 (64.20)	1300 (20.19)	5434 (84.39)	

* Kruskal Wallis test; # Pearson’s chi-squared test; GMC: geometric mean concentration.

**Table 4 vaccines-10-01806-t004:** Seroprevalence and tetanus antibody level in different doses.

Subgroup		*n*	GMC [IU/mL, (95% CI)]	*Z, p*	Proportion [*n* (%)]			Positive Rate (≥0.01 IU/mL) [*n* (%)]	*χ*^2^, *p*
					<0.01 IU/mL	0.01–0.1 IU/mL	≥0.1 IU/mL		
Doses	1	56	0.03	688000 *, <0.01	4 (7.14)	45 (80.36)	7 (12.50)	52 (92.86)	13.799 #, <0.01
	2	12	0.04		3 (25.00)	7 (58.33)	2 (16.67)	9 (75.00)	
	3	143	0.06		15 (10.49)	76 (53.15)	52 (36.36)	128 (89.51)	
	4	814	0.10		39 (4.79)	372 (45.70)	403 (49.51)	775 (95.21)	
	5	1039	0.10		74 (7.12)	430 (41.39)	535 (51.49)	965 (92.88)	
	Total	2064	0.09		135 (6.54)	930 (45.06)	999 (48.40)	1929 (93.46)	

* Jonckheere-Terpstra test, # Pearson’s chi-squared test; GMC: geometric mean concentration.

**Table 5 vaccines-10-01806-t005:** Seroprevalence and antibody level over time following basic and booster immunization of tetanus-toxoid-containing vaccines (TTCVs).

Interval	*n*	GMC [IU/mL, (95% CI)]	*Z, p*	Proportion [*n* (%)]			Positive Rate (≥0.01 IU/mL) [*n* (%)]	*χ*^2^, *p*
				<0.01 IU/mL	0.01–0.1 IU/mL	≥0.1 IU/mL		
0-	134	0.31	Z = 140200 *, <0.01	3 (2.24)	26 (19.40)	105 (78.36)	131 (97.76)	54.136 #, <0.01
1-	137	0.18		0 (0.00)	41 (29.93)	96 (70.07)	137 (100.00)	
2-	114	0.16		2 (1.75)	34 (29.82)	78 (68.42)	112 (98.25)	
3-	71	0.15		2 (2.82)	27 (38.03)	42 (59.15)	69 (97.18)	
4-	93	0.11		4 (4.30)	38 (40.86)	51 (54.84)	89 (95.70)	
5-	75	0.08		4 (5.33)	39 (52.00)	32 (42.67)	71 (94.67)	
6-	60	0.06		11 (18.33)	28 (46.67)	21 (35.00)	49 (81.67)	
7-	63	0.06		8 (12.70)	27 (42.86)	28 (44.44)	55 (87.30)	
8-	35	0.07		4 (11.43)	19 (54.29)	12 (34.29)	31 (88.57)	
9-	48	0.06		3 (6.25)	29 (60.42)	16 (33.33)	45 (93.75)	
10-	160	0.03		30 (18.75)	96 (60.00)	34 (21.25)	130 (81.25)	
Total	990	0.10		71 (7.17)	404 (40.81)	515 (52.02)	990 (92.83)	

* Jonckheere–Terpstra test; # Linear-by-Linear Association; GMC: geometric mean concentration.

**Table 6 vaccines-10-01806-t006:** Seroprevalence and tetanus antibody level since the last dose of tetanus-toxoid-containing vaccines (TTCVs).

Interval	*n*	GMC [IU/mL, (95% CI)]	*Z, p*	Proportion [*n* (%)]			Positive Rate (≥0.01 IU/mL) [*n* (%)]	*χ*^2^, *p*
				<0.01 IU/mL	0.01–0.1 IU/mL	≥0.1 IU/mL		
0-	263	0.23	629900 *, <0.01	7 (2.66)	68 (25.86)	188 (71.48)	256 (97.34)	79.598 #, <0.01
1-	296	0.13		6 (2.03)	118 (39.86)	172 (58.11)	290 (97.97)	
2-	250	0.15		4 (1.60)	88 (35.20)	158 (63.20)	246 (98.40)	
3-	212	0.11		7 (3.30)	99 (46.70)	106 (50.00)	205 (96.70)	
4-	186	0.11		7 (3.76)	84 (45.16)	95 (51.08)	179 (96.24)	
5-	103	0.10		7 (6.80)	44 (42.72)	52 (50.49)	96 (93.20)	
6-	79	0.07		11 (13.92)	37 (46.84)	31 (39.24)	68 (86.08)	
7-	85	0.07		9 (10.59)	34 (40.00)	42 (49.41)	76 (89.41)	
8-	64	0.07		5 (7.81)	36 (56.25)	23 (35.94)	59 (92.19)	
9-	71	0.07		4 (5.63)	41 (57.75)	26 (36.62)	67 (94.37)	
10-	301	0.04		50 (16.61)	171 (56.81)	80 (26.58)	251 (83.39)	
20-	66	0.03		9 (13.64)	50 (75.76)	7 (10.61)	57 (86.36)	
Total	1976	0.10		126 (6.38)	870 (44.03)	980 (49.60)	1850 (93.62)	

* Jonckheere-Terpstra test; # Linear-by-Linear Association; GMC: geometric mean concentration.

## Data Availability

The data supporting the findings of this study are contained within the article.

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
