# Peer review of "Seroepidemiology of Tetanus among Healthy People Aged 1–59 Years Old in Shaanxi Province, China"

_vaccines, 2022, doi:10.3390/vaccines10111806_

Round 1

Reviewer 1 Report

I was invited to revise the paper entitled "Seroepidemiology of tetanus among healthy people aged 1-59 years old in Shaanxi province, China". It was a cross-sectional study aimed to evaluate the seroprevalence of antibodies against tetanus among healthy people from a low-income Region of China.

The topic is interesting but the paper presents some limitations.

Major observations:

- Statistical analysis section was totally missing;

- Why did Authors analyzed age categories in this way? Why were the first ten years considered as single age class? it can bias the statistical analysis;

- Authors should present the baseline characteristics of all included patients;

- Sample size estimation was totally lacking;

- In teory, adults should perform a booster dose every 10 years. Why did Authors not take in account this point in the analysis? It was only poorly discussed in the late part of discussions.

Minor observations:

- Figure 1 has low quality;

- In introduction, TTCVs was not priorly defined;

Author Response

Response to Reviewer 1 Comments

Point 1: Statistical analysis section was totally missing;

Response 1: Thank you very much for your advice! I am very sorry for that the statistical analysis section was totally missing. The statistical analysis section was included in the original manuscript, but it was lost due to our negligence, when we use the Microsoft Word template. Now it has been added.

Point 2: Why did Authors analyzed age categories in this way? Why were the first ten years considered as single age class? it can bias the statistical analysis;

Response 2: Thank you very much for asking such a professional question! It was true that our age groups were not very reasonable. At your suggestion, we have readjusted the age groups according to the Chinese immunization program procedure and related published articles, namely, 1-2 years old group, 3-4 years old group, 5-6 years old group, 7-9 years old group, 10-19 years old group, 20-29 years old group, 30-39 years old group, 40-49 years old group, and 50-59 years old group.

Point 3: Authors should present the baseline characteristics of all included patients;

Response 3: Thank you very much for your advice! According to your suggestion, the baseline characteristics of all included patients including sex, age, ethnicity and regional distribution have been presented. For subjects over 18 years old, occupation and education level have been presented, too.

Point 4: Sample size estimation was totally lacking;

Response 4: Thank you very much for your careful advice. In the article, our description on sampling strategy is indeed not detailed enough in the “Materials and Methods” section. We have added the corresponding content in the manuscript. At the same time, we also introduce you to the outline of the sampling method, and the detailed scheme has been uploaded as an annex.

A large-scale epidemiological investigation of hepatitis B virus (HBV) was conducted in Shaanxi province (with a population of 38.35 million) by the Center for Disease Control and Prevention (CDC) in 2017. The overview of the sampling method of HBV investigation in Shaanxi province was as follows:

Survey objects and sampling methods

Survey objects were residents (living ≥ 6 months) aged 1 to 59 years old from 30 county-level settings in 10 prefectures of Shaanxi province. The sampling population was divided into three age groups: 1-14 years old, 15-29 years old and 30-59 years old.

  1. Sampling design

Stratified two-stage cluster random sampling method was used to extract the permanent population of different age groups in the whole population in 30 county-level disease surveillance sites of 10 prefecture-level cities in Shaanxi province. It is divided into six layers according to northern Shaanxi, central Shaanxi, southern Shaanxi, urban and rural areas. Each layer of each age group sample size equal capacity allocation.

  1. Sample size determination

The minimum sample size was determined according to the estimated prevalence of HBsAg seroepidemiological survey for HBV in the 2006 in Shaanxi province. The required sample size under simple random sampling was estimated, and then multiplied by the design effect (deff) coefficient to determine the actual sampling design. Since the results of HBV seroepidemiological survey in 2006 showed that the prevalence of HBsAg was 2.8 % (95 % CI: 2.66%-2.94%) in northern Shaanxi, 2.44 % (95 % CI: 2.32%-2.56%) in central Shaanxi and 4.9% (95 % CI: 4.66% -5.14% ) in southern Shaanxi, there was no significant difference among different regions. Therefore, the estimated prevalence of HBsAg in the permanent population aged 1-14, 15-29 and 30-59 in Shaanxi was used as the calculation marker to determine the minimum sample size required to determine the difference in the prevalence of HBsAg among different age groups. The minimum sample size required for the 1-14, 15-29 and 30-59 age groups was 1902, 2212 and 2737, respectively.

The above content is an overview of the sampling method of HBV investigation in Shaanxi province, and the detailed scheme has been uploaded as an annex.

In this study, in order to get a large enough sample, serums with a volume of more than 100 μL were selected to determine the seroprevalence of antibodies against tetanus among healthy people aged 1-59 years old in Shaanxi province. All the serum samples used for HBV serosurvey were stored at -20 °C before analysis. At last, a total of 6439 patients were included in this study.

Point 5: In theory, adults should perform a booster dose every 10 years. Why did Authors not take in account this point in the analysis? It was only poorly discussed in the late part of discussions.

Response 5: Thank you very much for your advice! Regarding this issue, the results section was indeed not detailed enough. According to your suggestions, we have added corresponding content in the results section 3.6 (3.6. Antibody level following basic and booster immunization of TTCVs) and 3.7 (3.7. Antibody level according to the interval since the last dose of TTCVs) to support the discussion section.

Point 6: Figure 1 has low quality;

Response 6: Thank you very much for your advice! Image quality and resolution have been improved.

Point 7: In introduction, TTCVs was not priorly defined;

Response 7: Thank you very much for your careful reading! We are very sorry for that TTCVs was not priorly defined in introduction section, and we have explained the abbreviation “TTCVs” with “tetanus-toxoid-containing vaccines” in introduction section.

Reviewer 2 Report

Estimated Authors of the paper Seroepidemiology of tetanus among healthy people aged 1-59 years old in Shaanxi province, China,

I've read with great interest your report, whose content is highly consistent with aims and topic covered by TropMed, also representing an interesting issue for international readers.

In this study, Zhang C et al. report on a serosurvey performed in various provinces of Mainland China, for a total of around 6,500 individuals. The serosurvey was originally designed for providing information on HBV (please provide the appropriate reference to the original study), and data on tetanus were derived from samples taken for this original study.

Briefly, the study is consistent with previous reports stressing the inappropriate vaccination coverage on Tetanus in several areas of mainland China, and with the failing coverage in older age groups resulting from both waning immunity and lack of baseline treatment.

From my P.O.V., the present study could be accepted for publication, but significant improvements are requested, and more precisely:

1) please discuss how the data were managed in the methods section, and more precisely describe the statistical and epidemiological analyses you've performed. For instance, we know that data were compared by means of the Jonckheere's trend test, but the rationale for this choice has not been provided. This is particularly important as an uncommon test. 

2) Authors should provide, at least as annex document, some information on the sampling strategy you did rely on. How were the sampled individuals choose on a vastly larger population? I mean: in Xi'an alone, the last census reports a total of 12,952,907 residents, but your analyses were performed on around 1400 participants, that is 1/10,000 individuals.

3) Tables are not properly designed: as a common habit, first you should report crude numbers, and then the percent estimates - while in your table these data are conversely reported. Moreover, tables should be understandable without relying on the main text, and your tables contain several acronyms and abbreviations that are not explained in the labels or captions, e.g. GMC and TTCV.

4) In table 1, the first column should be refined: categories are, I presume, 1,2,3,4,5,6,7,8,9,10-19, 20-29, 30-39 and so on, but the text is seemly cut off from the table. 

5) please enlarge figure 1

6) please provide a map reporting the areas involved in the analysis

Author Response

Response to Reviewer 2 Comments

Point 1: The serosurvey was originally designed for providing information on HBV (please provide the appropriate reference to the original study), and data on tetanus were derived from samples taken for this original study.

Response 1: Thank you very much for your advice! The large-scale epidemiological investigation of hepatitis B virus (HBV) was conducted in Shaanxi province (with a population of 38.35 million) by the Center for Disease Control and Prevention (CDC) in 2017. So far, all work of this project including experiments, data analysis and so on, has been completed. However, unfortunately, no relevant articles have been published. In order to make you understand our work in detail, the file “the project plan on HBV seroepidemiological survey in Shaanxi province” has been uploaded as an attachment, including survey purposes, survey subjects and sampling methods, experimental methods and data analysis and so on.

Point 2: please discuss how the data were managed in the methods section, and more precisely describe the statistical and epidemiological analyses you've performed. For instance, we know that data were compared by means of the Jonckheere's trend test, but the rationale for this choice has not been provided. This is particularly important as an uncommon test.

Response 2: Thank you very much for your advice! I am very sorry for that the statistical analysis section was totally missing. The statistical analysis section was included in the original manuscript, but it was lost due to our negligence, when we use the Microsoft Word template. Now it has been added.

The results of IgG against tetanus for 6, 439 subjects show non-normal distribution through Test of Normality (Kolmogorov-Smirnov) by SPSS software, so relevant tests were performed by non-parametric test such as Kruskal-Wallis test or Jonckheere-Terpstra test. The GMC of antibodies were compared among groups using Kruskal-Wallis test or Jonckheere-Terpstra test, and seropositivity was compared between subgroups using chi-squared. Linear-by-Linear Association was used to test the change trend. SPSS software (version 25.0; SPSS Inc., Chicago, IL, USA) was performed for data analysis and a P-value of <0.05 was considered significant.

According to your suggestion, the relevant content of the original manuscript has been described in more detail and has been added.

Point 3: Authors should provide, at least as annex document, some information on the sampling strategy you did rely on. How were the sampled individuals choose on a vastly larger population? I mean: in Xi'an alone, the last census reports a total of 12,952,907 residents, but your analyses were performed on around 1400 participants, that is 1/10,000 individuals.

Response 3: Thank you very much for your careful advice. In the article, our description on sampling strategy is indeed not detailed enough in the “Materials and Methods” section. We have added the corresponding content in the manuscript. At the same time, we also introduce you to the outline of the sampling method, and the detailed scheme has been uploaded as an annex.

A large-scale epidemiological investigation of hepatitis B virus (HBV) was conducted in Shaanxi province (with a population of 38.35 million) by the Center for Disease Control and Prevention (CDC) in 2017. The overview of the sampling method of HBV investigation in Shaanxi province was as follows:

Survey objects and sampling methods

Survey objects were residents (living ≥ 6 months) aged 1 to 59 years old from 30 county-level settings in 10 prefectures of Shaanxi province. The sampling population was divided into three age groups: 1-14 years old, 15-29 years old and 30-59 years old.

  1. Sampling design

Stratified two-stage cluster random sampling method was used to extract the permanent population of different age groups in the whole population in 30 county-level disease surveillance sites of 10 prefecture-level cities in Shaanxi province. It is divided into six layers according to northern Shaanxi, central Shaanxi, southern Shaanxi, urban and rural areas. Each layer of each age group sample size equal capacity allocation.

  1. Sample size determination

The minimum sample size was determined according to the estimated prevalence of HBsAg seroepidemiological survey for HBV in the 2006 in Shaanxi province. The required sample size under simple random sampling was estimated, and then multiplied by the design effect (deff) coefficient to determine the actual sampling design. Since the results of HBV seroepidemiological survey in 2006 showed that the prevalence of HBsAg was 2.8 % (95 % CI: 2.66%-2.94%) in northern Shaanxi, 2.44 % (95 % CI: 2.32%-2.56%) in central Shaanxi and 4.9% (95 % CI: 4.66% -5.14% ) in southern Shaanxi, there was no significant difference among different regions. Therefore, the estimated prevalence of HBsAg in the permanent population aged 1-14, 15-29 and 30-59 in Shaanxi was used as the calculation marker to determine the minimum sample size required to determine the difference in the prevalence of HBsAg among different age groups. The minimum sample size required for the 1-14, 15-29 and 30-59 age groups was 1902, 2212 and 2737, respectively.

The above content is an overview of the sampling method of HBV investigation in Shaanxi province, and the detailed scheme has been uploaded as an annex.

In this study, in order to get a large enough sample, serums with a volume of more than 100 μL were selected to determine the seroprevalence of antibodies against tetanus among healthy people aged 1-59 years old in Shaanxi province. All the serum samples used for HBV serosurvey were stored at -20 °C before analysis. At last, a total of 6439 patients were included in this study.

Point 4: Tables are not properly designed: as a common habit, first you should report crude numbers, and then the percent estimates - while in your table these data are conversely reported. Moreover, tables should be understandable without relying on the main text, and your tables contain several acronyms and abbreviations that are not explained in the labels or captions, e.g. GMC and TTCV.

Response 4: Thank you very much for your advice! Indeed, as you mentioned that “as a common habit, first you should report crude numbers, and then the percent estimates-while in your table these data are conversely reported.”, and we have re-designed all the tables in this study. Now, the crude numbers are before the percent estimates.

Moreover, according to your suggestion, in order to make tables be understandable without relying on the main text, the acronyms and abbreviations such as GMC and TTCV e.g. have been explained in the captions below each table.

Point 5: In table 1, the first column should be refined: categories are, I presume, 1,2,3,4,5,6,7,8,9,10-19, 20-29, 30-39 and so on, but the text is seemly cut off from the table.

Response 5: Thank you very much for your careful advice. According to your suggestion, we have refined the first column in table 1, and the text part has also been adjusted accordingly.

Point 6: please enlarge figure 1

Response 6: Thank you very much for your advice! Image quality and resolution have been improved.

Point 7: please provide a map reporting the areas involved in the analysis

Response 7: Thank you very much for your advice! A map (Figure 1) reporting the areas involved in the analysis have been provided in section 2.1(2.1. Subjects and study design).

Round 2

Reviewer 1 Report

Authors addressed all comments. In my opinion the paper can be accepted for publication.

Author Response

Thank you very much for your support to our work!

Reviewer 2 Report

Estimated Authors,

the paper has been extensively improved, and most of my concerns were properly removed.

I've only a minor request before guaranteeing a full acceptance.

That is:

please be aware that the reference [13] is substantially out of date; I would suggest to replace it with the following ones:

https://pubmed.ncbi.nlm.nih.gov/24370712/

https://pubmed.ncbi.nlm.nih.gov/27251030/

As such amendments can be done in the pre-publishing editorial stage, I'm endorsing the acceptance for this publication.

Author Response

Thank you very much for your advice! We have read the two articles you recommended and we think your suggestions are reasonable! According to your suggestion, the original reference has been replaced by these two articles! And the content of the original text has also been adjusted (the third paragraph of the introduction section)!